# microRNA, a Subtle Indicator of Human Cytomegalovirus against Host Immune Cells

**DOI:** 10.3390/vaccines10020144

**Published:** 2022-01-19

**Authors:** Mengyao Yu, Yuexinzi Jin, Shichang Zhang, Jian Xu, Jiexin Zhang

**Affiliations:** 1Department of Laboratory Medicine, The First Affiliated Hospital with Nanjing Medical University, Nanjing 210029, China; ymy1994@njmu.edu.cn (M.Y.); jinyuexinzi@njmu.edu.cn (Y.J.); zsc78@njmu.edu.cn (S.Z.); 2Branch of National Clinical Research Center for Laboratory Medicine, Nanjing 210029, China

**Keywords:** HCMV, microRNA, innate immunity, adaptive immunity, latency

## Abstract

Human cytomegalovirus (HCMV) is a double-stranded DNA virus that belongs to the β-herpesvirus family and infects 40–90% of the adult population worldwide. HCMV infection is usually asymptomatic in healthy individuals but causes serious problems in immunocompromised people. We restricted this narrative review (PubMed, January 2022) to demonstrate the interaction and molecular mechanisms between the virus and host immune cells with a focus on HCMV-encoded miRNAs. We found a series of HCMV-encoded miRNAs (e.g., miR-UL112 and miR-UL148D) are explicitly involved in the regulation of viral DNA replication, immune evasion, as well as host cell fate. MiRNA-targeted therapies have been explored for the treatment of atherosclerosis, cardiovascular disease, cancer, diabetes, and hepatitis C virus infection. It is feasible to develop an alternative vaccine to restart peripheral immunity or to inhibit HCMV activity, which may contribute to the antiviral intervention for serious HCMV-related diseases.

## 1. Introduction

Human cytomegalovirus (HCMV) is a β-herpesvirus with double-stranded DNA, and it has the largest genome (approximately 230 kb) among its family members. In developed countries, 40–60% of adults are HCMV infected, and the seroprevalences in some developing countries approach 100% [1,2]. The pathophysiological process of HCMV invasion in the human body causes a multilayered cascade of immune reactions, as follows: it aims at various signaling transduction pathways involving the whole peripheral immune system, leading to the impaired antigen recognition ability of CD8^+^ T cells, the abnormal differentiation of CD4^+^ T cells, the inhibition of the natural killer (NK) cell killing effect, or the reduction in proinflammatory cytokine secretion, rather than a single immune cell or a specific immune molecule. In addition, the immunosuppression status caused by HCMV infection may lead to a secondary infection by other pathogens and further deteriorate the host’s immune function.

Upon HCMV infection, it first contacts the cell membrane, then fuses with the cell membrane and enters the cytoplasm. The nucleocapsid containing viral DNA enters the nucleus through the activities of a series of viral proteins. At this time, if the HCMV genome can initiate transcription of the immediate early (IE) genes by the main immediate early promoter (MIEP), the newly assembled progeny virus will lyse the host cell and release to establish a lytic infection. The HCMV genome encodes a variety of regulatory proteins (such as IE2, IE72, and glycoproteins) that regulate viral DNA replication and virus offspring propagation [3,4]. In addition, the proteins maintaining its structure (such as viral tegument protein phosphoprotein 65 (pp65), glycoprotein B, and glycoprotein H) also play indispensable roles in the regulation of viral transcription, the cell cycle, and even the immune response [5,6]. Otherwise, HCMV infection will be latent, which is very important for virus long-term survival [7].

HCMV latent infection, defined by the maintenance of viral genomes in the absence of new virus production, causes serious clinical symptoms in immunocompromised individuals, such as those undergoing solid organ or hematopoietic stem cell transplants. These patients are susceptible to HCMV reactivation from latency and virus replication in numerous tissues and organs, causing significant morbidity and mortality [8]. To date, the treatment of latent HCMV infection is still a dilemma because all currently optional therapies are designed for viruses during replication [9]. Thus, it is important to understand the pathological mechanisms of HCMV latency and to clarify a series of targets that may be involved in the conversion process to prevent serious cytomegalovirus disease.

Viruses encode short noncoding RNAs (microRNAs, miRNAs) that cooperate with viral proteins to regulate the expression of viral and host genes. MiRNAs are long noncoding RNAs (measuring ~22 nucleotides) that are encoded by eukaryotes and viruses (Figure 1). They are incorporated into the multiprotein RNA-induced silencing complex (RISC). RISC leads to the transcriptional repression of messenger RNAs (mRNAs) through complementarity between nucleotides 2–8 of the miRNAs (the seed region) and 3′-untranslated regions (3′-UTR) of the mRNAs [10,11,12]. The first miRNA was identified by Lee and colleagues in Caenorhabditis elegans in 1993 [13]. Pfeffer first identified miRNAs in Epstein–Barr virus (EBV) [14]. More than 200 virus-encoded miRNAs have been identified in double-stranded DNA viruses, of which 26 HCMV-encoded miRNAs and their potential targets have been reported to play various roles in viral replication, immune evasion, and host-cell fate [15,16]. Interestingly, in contrast to other herpes viruses, genes coding miRNAs are scattered throughout HCMV genome which implies each HCMV-encoded miRNA may be regulated by its own regulatory sequence [16]. Here, we will discuss in detail latent HCMV-encoded miRNAs and their validated targets to address their dynamic interactions with host immune cells.

## 2. Host Immune Response Overview to HCMV Infection

### 2.1. Innate Immune Responses

The innate immune response is an early defense mechanism and is largely initiated by pattern recognition receptors (PRRs) upon the sensing of HCMV. It triggers multiple signaling pathways to produce tumor necrosis factor (TNF)-α, interferon (IFN)-β, interleukin (IL)-6 and other proinflammatory cytokines to attract immune cells to the site of infection [17,18,19]. Among the PRRs, Toll-like receptors (TLRs) are the most extensively studied and are largely expressed by antigen-presenting cells (APCs), such as monocytes, macrophages, dendritic cells (DCs), and NK cells. These immune cells specialize in sensing pathogens and producing cytokines. APCs not only have direct antiviral properties but can also promote an adaptive immune response [17].

Monocytes are generally considered to be a major site where HCMV maintains latency and spreads all over. Following primary infection, infected monocytes carry HCMV particles to nonselective tissues and organs to cause an initial immune contact and response [20]. Viral particles specifically inhibit the surface expression of constitutive and inflammatory chemokine receptors on infected monocytes, which hampers the chemokine-driven migration of monocytes and the ability to recruit other immune cells. Thus, HCMV can be transferred from infected monocytes to uninfected cells [20]. It has been reported that the clinical HCMV strain can reprogram human hematopoietic progenitor cells (HPCs) into a unique monocyte subset necessary for establishing latency by activating STAT3. This monocyte subset expresses higher levels of the coinhibitory molecules B7-H4, IL-10, inducible nitric oxide synthase (iNOS), and NO. Excess intracellular NO suppresses HCMV IE expression and viral replication, leading to successful immunosuppression and HCMV latency [21].

Nevertheless, massive immune responses in the host will be triggered to protect against HCMV infection, especially through IFN type I (IFN-I) [22]. Monocyte-derived macrophages (moMΦs) or dendritic cells (moDCs) present strong IFN-I responses by HCMV sensing in a TLR 9- or cyclic GMP/AMP synthase (cGAS)-dependent manner, respectively [23]. A recent report suggested that monocyte-derived cells only expressed IFN-I when stimulated by cell-free HCMV or upon encountering HCMV-infected cells that already produced cell-free virus. Upon coculture of infected epithelial/endothelial cells and moMΦs or moDCs, antiviral responses were induced to limit HCMV spread, which was dependent on cell–cell contact. In addition, cell-free supernatants from coculture experiments also inhibited virus spread, implying that soluble factors were critically needed. Interestingly, this antiviral effect was independent of IFN-γ, TNF-α, and IFN-I [22].

DCs are the most typical APCs. They activate native T cells and play an important role in the process of inducing T cell activation or tolerance by presenting antigens to lymphocytes via MHC I and MHC II molecular pathways [24]. Reeves and colleagues observed that latent HCMV can be reactivated during monocyte differentiation into DCs by IL-6 and downstream extracellular signal-regulated kinase-mitogen-activated protein kinase (ERK-MAPK) signaling [25,26]. HCMV-infected monocytes block the cytokine-induced differentiation of monocytes into functionally active DCs [17]. The following two main subtypes of DCs have been identified in the blood: plasmacytoid dendritic cells (pDCs) and myeloid DCs (mDCs). HCMV-infected cells can directly trigger pDCs to enhance the antiviral IFN-I response even in the absence of cell-free virus [22]. PDCs produce high amounts of IFN-α, exhibit increased survival, and decreased sensitivity upon HCMV exposure [27]. They also secrete IL-6 and IL-10 upon HCMV infection through engagement with the TLR7 and/or TLR9 pathways. Although HCMV in pDCs has an inhibitory effect on T-cell proliferation through IL-12 [28], it triggers B-cell activation, proliferation, and antibody production [29].

NK cells are an important component of the innate immune system to fight against HCMV infection. Studies have found multiple genetic loci (e.g., UL16 and UL142; UL = unique long) in the HCMV genome, and their transcripts inhibit NK cell recognition [30]. Natural killer Group 2 member D (NKG2D) belongs to the C-type lectin-like activating receptor family that is expressed ubiquitously on all NK cells to mediate immune surveillance on virus-infected cells after binding with NKG2D ligands (NKG2DLs). The HCMV UL16- and UL142-encoding proteins have high affinity for NKG2DLs and retain them in the cis Golgi other than on the cell surface of the infected cells to avoid NK cell recognition [31]. In this context, stress-induced cellular ligands MIC-A, MIC-B, and ULBPs, which are recognized by the activating receptor NKG2D, will be overexpressed upon viral infection, and they are central in NK-mediated immune responses against HCMV [32]. A significant increase in NK cells has been found to occur in the early stage of primary infection, in congenital infected neonates and in transplant recipients [33]. Activated NK cells produce proinflammatory cytokines (e.g., TNF-α and IFN-γ) and cytotoxic granules containing effector molecules (e.g., granzyme B and perforin) to cause the lysis or apoptosis of infected cells [34]. NK cells also directly kill transformed and infected cells via antibody-dependent cellular cytotoxicity (ADCC) [35,36]. Other non-HLA-I-specific activating receptors also play a role against HCMV. For example, NKp30, a natural cytotoxicity receptor (NCR), participates in the recognition and killing of HCMV-infected cells [37]. Similar to NKG2D and NCRs, the activating coreceptor DNAM1 recognizing PVR and Nectin-2 (CD112) plays a role against HCMV, as demonstrated by different evasion strategies reducing DNAM-1 signaling [38,39]. Importantly, NK cells also contribute to early viral defense by recognizing envelope glycoproteins from HCMV virions via TLR2 to produce IFN-γ, further promoting antiviral immune responses [40]. It has been confirmed that HCMV-derived cytokines augment NK cell cytotoxicity vice visa. An HCMV gene (UL111a) encodes cmvIL-10, a virokine homologous to human IL (hIL)-10, which substantially enhances NK cell cytotoxicity through the natural cytotoxicity receptors NKp30 and NKp46 and through the C-type lectin-like receptors NKG2C and NKG2D. Antibody-dependent cell-mediated cytotoxicity triggered by CD16 also increased significantly with short-term cmvIL-10 exposure [41].

### 2.2. Adaptive Immune Response

The control of HCMV viral replication and viral spreading is mediated by adaptive immune responses, particularly the T cell response [42]. As HCMV infection progresses, CD4^+^ and CD8^+^ T cells recognize viral peptides presented by MHC-II and MHC-I molecules, respectively, and become major effectors of the adaptive immune response. Activated CD4^+^ T cells present a typical T helper (Th) 1 cytokine secretion signature and become capable of secreting large amounts of IL-2, IFN-γ, and TNF-α to promote the killing effect of CD8^+^ T cells and to assist B cells in producing antibodies. In addition to these effector cytokines, Th1 CD4^+^ T cells express perforin and granzyme B, which mediate the lysis of HCMV-infected cells [43]. HCMV-specific CD8^+^ T cells lyse infected cells that present HCMV peptides on MHC-I through the secretion of granzyme B, perforin, and IFN-γ within hours of stimulation [44,45]. HCMV-specific CD8^+^ T cells also produce ligands such as C-C motif chemokine ligands (CCL4 and CCL5) to recruit monocytes, macrophages, NK cells, and activated T cells to amplify local inflammation [46]. However, the US2–US11 (US = unique short) region of the HCMV genome is specifically dedicated to interfering with the presentation of antigenic peptides to CD8^+^ T cells [47].

HCMV elicits adaptive NK cells. Infected host cells trigger the expansion of NKG2C^+^ NK cell subsets with a high capacity for granzyme B, IFN-γ, and TNF-α production even during latent infection [48]. Together with CD94 and NKG2C, they form a heterodimeric NK cell activating receptor to transduce signals via the adaptor protein DAP12 by binding its ligand HLA-E [49]. NKG2C^+^ NK cell expansion is further supported by CD2 activation and IL-12. Adaptive NK cells exhibit the downregulation of the transcription factor PLZF, which regulates the expression of the signaling adaptors FcRγ, SYK, and EAT-2, and these changes may indicate adaptive NK cells for enhanced activation and effector function during HCMV infection [50].

NKT cells are innate lymphoid cells derived from the thymus. They recognize lipids and glycolipid antigens presented by CD1d [51]. Invariant NKT (iNKT) cells are the major subset of NKT cells, expressing both NK cell receptors and invariant T cell receptors [52]. It has been reported that iNKT cells have an additional immunoregulatory role of bridging innate and adaptive immunity by receiving glycolipids delivered from CD1d molecules and immediately releasing large quantities of cytokines, such as IL-4 and IFN-γ. HCMV encodes US2, which interacts with CD1d and leads to ubiquitin-dependent proteasomal degradation to inhibit the antigen-presenting ability of MHC-Ia and MHC-II and thus escape immune surveillance by T cells [53].

## 3. The Regulation Mechanisms of Latent HCMV on Host Immune Cells

First, HCMV-encoded miRNAs can escape immune surveillance in a nonimmunogenic manner to favor intracellular latency. They reduce viral replication by downregulating IE expression or by silencing major viral proteins, or they suppress viral particle formation by targeting multiple host genes related to cell cycle control (e.g., cyclin E2 (CCNE2), collagenase stimulatory factor (CD147), BRCA1/BRCA2-containing complex, subunit 3 (BRCC3), EP300 interacting inhibitor of differentiation 1 (EID1), microtubule-associated proteins, RP/EB family member 2 (MAPRE2), and histone proteins (H3F3B)) without recognition by host immune cells [54]. More importantly, HCMV-encoded miRNAs can target signaling pathways involved in cell apoptosis, immune response, and proinflammatory cytokine production to avoid elimination by the host and to achieve immune evasion. However, virus activation is inevitable when the host immune system is compromised or when obligatory differentiation stimuli occur, which initiates instantaneous host damage and fatal consequences (Table 1). The importance of HCMV-encoded miRNAs is only beginning to be elucidated.

During the latent stage, HCMV encodes miRNAs to inhibit its own DNA synthesis in infected cells and to reduce the production of inflammatory mediators synthesized by host cells mainly through directly suppressing their transcription or interfering with the signaling pathways that stimulate their production, thereby enabling long-term coexistence in host cells. Moreover, HCMV-encoded miRNAs also blunt the host immune response by weakening immune cells (Figure 2).

### 3.1. HCMV-Encoded miRNAs Inhibit Viral DNA Replication

HCMV genomes exist as episomes in the nucleus after infecting host cells. Viral gene expression is mainly classified into IE and early (E) and late (L) gene expression. MIEP initiates the transcription of the IE1 and IE2 genes, which are the principal transcriptional activators necessary for virus early gene expression and play essential roles in immune evasion as well as in viral replication [88,89]. Early genes are involved in viral DNA replication and the subsequent activation of late gene expression, which plays essential roles in capsid assembly and virion release [90]. MiR-UL112 is an important HCMV-encoded miRNA that specifically binds to the 3′-UTRs of IE mRNAs, which acts as a part of the strategy for HCMV to enter host cells and maintain its latency by host immune evasion [91]. For example, miR-UL112-1 has been shown to target the IE72 transcript and attenuate its expression to decrease viral DNA multiplication [92]. Compared with wild-type HCMV strain-infected cells, HCMV knockout of miR-UL112-1 significantly induced IE72 expression during the latent infection of monocytes [4,91]. Other potential targets of miR-UL112 include UL112/113 and UL120/121. Interestingly, both IE72- and UL112/113-encoded products are important for viral replication. Although the biological function of UL120/121 is still elusive, some researchers have suggested that UL120/121 might encode exons within MIE family transcripts [92]. An additional target for miR-UL112 is UL114 (the viral uracil DNA glycosylase), which is encoded on the strand antisense to miR-UL112. UL114 deregulation by miR-UL112 hinders the virus from properly excising uracil residues from viral DNA and fails to control virus replication [56].

HCMV-encoded miR-US25-1 and miR-US25-2 were highly expressed during latent infections to inhibit viral replication. MiR-US25-1-5p was confirmed to inhibit HCMV DNA replication during infection by directly interacting with a series of cellular genes, including tyrosine 3-monooxygenase/tryptophan 5-monooxygenase activation protein epsilon (YWHAE), ubiquitin B (UBB), phosphoprotein B23 (NPM1), and eat shock protein 90 kDa alpha, class A member 1 (HSP90AA1) [93]. MiR-US25-1 can bind target sites that are primarily located within the 5′UTRs of genes associated with cell cycle control, including cyclin E2, BRCC3, EID1, MAPRE2, and CD147 [58]. A study conducted by Stern-Ginossar demonstrated that the overexpression of miRNA-US25-1 and miRNA-US25-2 in human foreskin fibroblasts (HFFs) blocks the expression of IE72 and pp65 through targeting 3′-UTR of their mRNAs, thus reducing the DNA synthesis and viral replication of HCMV [56]. Qi et al. [59] reported that miR-US25-2-3p directly inhibits the expression of eukaryotic translation initiation Factor 4A1 (eIF4A1), leading to a decrease in HCMV DNA synthesis.

Other HCMV-encoded miRNAs reported to mediate viral DNA replication during latent infection are miR-US33-5p, miR-UL148D, miR-UL36, and miR-US5-1. Syntaxin3 (STX3), a component of soluble N-ethylmaleimide-sensitive factor attachment protein receptors (SNAREs), participates in membrane-fusion events and regulates cell growth and cytokinesis. It was found to be a direct target of miR-US33-5p [60]. STX3 is incorporated into the HCMV viral envelope, and its depletion alters viral particle formation [94]. MiR-UL148D is highly expressed during the late stages of experimental latent HCMV infection in CD34^+^ progenitor cells and in Kasumi-3 cells. This miRNA inhibits the expression of immediate early response gene 5 (IER5) by directly targeting the 3’-UTR of IER5 mRNA, followed by the enhanced expression of the cell division cycle 25B (CDC25B) protein [57]. CDC25B can activate cyclin-dependent kinase-1 (CDK-1) to suppress HCMV IE1 and lytic gene transcription [95]. MiR-UL36 is a viral miRNA contributing to HCMV replication through targeting the viral UL138 gene, which has previously been identified as a novel latency-associated determinant of HCMV infection [81]. MiR-US5-1 has been shown to directly target the DNA replication inhibitor geminin, and its overexpression competitively inhibits viral replication and stimulates DNA synthesis in human brain glioma cells (U373) [82]. Taken together, these findings indicate that HCMV-encoded miRNAs target both viral genes and host genes to control viral DNA replication and to maintain viral latency.

### 3.2. HCMV-Encoded miRNAs Regulate Biological Functions of Host Immune Cells

Apoptosis is an antiviral defense mechanism to eliminate virus-infected cells and restrict the spread of virus and prevent persistent infection [70]. HCMV upregulates antiapoptotic miRNAs, including miR-UL-70-3p, miR-UL36-5p, miR-UL148D, miR-US5-1, and miR-UL112-3p, to inhibit apoptosis of host cells. MiR-UL70-3p binds to the 3′-UTR of modulator of apoptosis 1 (MOAP1) gene, which encodes the MOAP1 protein to inhibit mitochondrial-dependent apoptosis by countering the stabilization of the proapoptotic protein Bax [70]. Guo et al. [71] found that miR-UL36-5p directly targets adenine nucleotide translocator 3 (ANT3) mRNA. The ANT3 protein is located on the mitochondrial inner membrane as a core component of the mitochondrial permeability transition pore (MPTP) to regulate apoptosis [96]. ANT3 expression was confirmed to be downregulated by miR-UL36-5p overexpression, which reduces apoptosis at an early stage of HCMV infection and provides a suitable cellular environment for virus replication. Endoplasmic reticulum-to-nucleus signaling 1 (ERN-1) is activated by sensing the presence of unfolded proteins in the ER and then mediates the ER stress response to eliminate them. However, it will cause autophagy and cell function deterioration if ER stress cannot be reversed [97]. The tumor suppressor putative HLA-DR-associated proteins (PHAP) promote caspase-9 activation, which contributes to apoptosome formation and further activates caspase-3. The activation of caspase-3 plays a crucial role in mitochondrial-induced apoptosis [98]. Babu et al. used a computational method and found that miR-UL148D targets both ERN-1 mRNA and PHAP mRNA to inhibit cell death [70]. Human immediate early gene X-1 (IEX-1) is a cell apoptosis regulator whose expression is inhibited by HCMV-encoded miR-UL148D [69]. It has been observed that miR-US5-1 and miR-UL112-3p are expressed in CD34^+^ hematopoietic progenitor cells (HPCs) during the early stages of HCMV infection. They act to reduce FOXO3a (a member of the mammalian Forkhead Box O family of transcription factors) activity and BCL2L11 expression to promote infected cell survival [99]. Interestingly, Kim et al. found that FAS is a target of miR-UL36-3p, miR-US5-1, and miR-US5-2-3p; Caspase-3 is downstream of miR-US25-2-3p, miR-112-5p, and miR-UL22A-5p; and Caspase-7 is a target of both miR-UL22A-3p and miR-US4-3p [61]. Therefore, HCMV-encoded miRNAs inhibit apoptosis processes, such as mitochondrial oxidative stress and ER stress, by regulating the expression of key proteins in the apoptotic process to counter cellular defenses. It is possible that apoptosis inhibition of immune cells, which leads to an increasing proportion of nonfunctional mature immune cells, contributes to virus immune escape. The mechanism between HCMV-encoded miRNAs and host immune cell apoptosis still needs to be discussed in detail.

It is known that HCMV can employ miRNAs to interfere with immune responses to survive. MICB is critical for the cytocidal function of NK cells toward virus-infected cells [63]. NKG2D is an activating receptor of NK cells that can activate NK cells by binding with its ligand MICB, resulting in the killing of target cells expressing MICB. MiR-UL112 silences MICB mRNA translation by binding to its 3′-UTR specifically, thereby decreasing MICB binding to NKG2D [62]. In addition to the MICB pathway, miR-UL112 also attenuates NK cell-mediated cytotoxicity by downregulating IL-32-, IFN-, and TLR2-mediated NF-κB signaling [74,100]. The overexpression of miR-UL112 in peripheral blood mononuclear cells (PBMCs) results in the downregulation of type I IFN on the NK cell membrane [100]. This is consistent with previous reports that the decrease in type I IFN partially inhibits the activation of NK cells [40]. Huang et al. found that, with the increase in miR-UL112-1 expression observed while HCMV infection is prolonged, IL-32 transcription and protein levels in HCMV-infected human lung fibroblast MRC-5 cells are significantly decreased, which may reduce the content of TNF-α and inhibit the activation of NK cells [101]. HCMV also stimulates the shedding of the NKG2D ligand MICA by reducing the expression of the endogenous inhibitor of metalloproteases tissue inhibitors of metalloproteinase 3 (TIMP3), which is modulated by miR-US25-2-3p [102]. Activin A is a member of the transforming growth factor-beta (TGF-β) ligand superfamily, and it upregulates IL-6 secretion to exert a proinflammatory effect through the participation of the activin A receptor ACVR1B [103]. Lau B et al. [104] confirmed that abundant miR-UL148D downregulates IL-6 secretion by blocking ACVR1B to prohibit NK cell activation in HCMV-infected human monocytes during latent infection [76]. MiR-UL148D directly induces the degradation of human T cell-secreted RANTES to escape the immune response [77]. RANTES is also a chemokine that induces the proliferation and activation of NK cells with the help of T cells to release particular cytokines (IL-2 and IFN-γ) [105]. The latest study showed that HCMV-miR-US33as-5p binds the 3′-UTR of IFN receptor 1 (IFNAR1) and interferes with the typical IFN signaling pathway to limit the release of IFN-stimulated genes (ISGs), which can encode antiviral proteins [68].

During HCMV latency, miR-UL112-1 ensures delicate control of IE72 expression so that it not only inhibits viral DNA replication but also prevents the recognition of infected cells by the host’s potent pre-existing antiviral cytotoxic T cell (CTL) response [91]. Kim et al. demonstrated that miR-US4-1 targets endoplasmic reticulum aminopeptidase 1 (ERAP1), a highly polymorphic key component of antigen processing, to inhibit MHC class I–mediated antigen presentation, resulting in recognition restriction of viral antigen by CD8^+^ T cells [47]. ERAP1 is an aminopeptidase involved in catalyzing antigenic peptide production in the ER and trimming long peptides to the lengths required for presentation by MHC class I [106]. In the RISC complex, miR-US4-1 targets the 3′-UTR of ERAP1 mRNA in a manner that destabilizes or degrades mRNA [47]. Whether ERAP1 is a real target has been debated because the original seed sequence of miR-US4 used by Kim et al. was not correct based on new deep sequencing data [72]. A recent study reported that HCMV miR-UL112-5p targets ERAP1, thereby inhibiting the processing and presentation of the HCMV pp65 peptide (495-503) to specific CTLs [65]. The relevance of ERAP1 SNPs within miRNA binding sites in modulating viral miRNA–mRNA interactions and the possible consequent individual susceptibility to HCMV infection has been reviewed elsewhere by Ombretta [107]. Overall, this new evidence indicates that targeting the ERAP1 gene can be a promising immune evasion strategy for HCMV.

### 3.3. HCMV-Encoded miRNAs Impair Host Inflammatory Signal Transduction Pathways

Inflammatory cascades are crucial for HCMV reactivation, and viruses make special efforts to inhibit the axis of cytokine synthesis and secretion. Cellular receptor-mediated signaling pathways are generally assumed to play an important role during the earliest immune response to virus infection. HCMV envelope glycoproteins B and H (gB and gH) interact directly with TLR2 [a major PRR trigger of nuclear factor kappa-B (NF-κB) signaling] on the plasma membrane to further stimulate the NF-κB pathway and release inflammatory mediators and cytokines (such as IL-6, TNF-α, and IFN-β) [74]. However, miR-UL112-3p downregulates TLR2 expression in fibroblasts and THP-1 cells through the inhibition of TLR2-induced interleukin-1 receptor-associated kinase (IRAK1, a kinase located upstream of NF-κB) activation, which significantly decreases proinflammatory cytokines [74]. In another study, miR-UL112-3p regulated TLR2, whose deletion was observed to significantly reduce proinflammatory secretions and to decrease NK cell populations in mouse models [4,108]. CD147 (or EMMPRIN) is a type-I transmembrane glycoprotein of the immunoglobulin superfamily. It was confirmed to play an essential role in HCMV-triggered early antiviral signaling [109]. As a ligand for CD147 and a proinflammatory cytokine, the expression and secretion of cyclophilin A (sCyPA) were upregulated in response to HCMV stimuli [110]. HCMV-encoded miR-US25-1-5p targets the 3′ UTR of CD147 mRNA and mediates HCMV-triggered antiviral signaling via the sCyPA-CD147-ERK (extracellular regulated protein kinases)/NF-κB axis signaling pathway, whereas CD147 knockdown significantly decreases the HCMV-induced activation of NF-κB and IFN-β [78]. MiR-UL148D targets cellular ACVR1B, a receptor of the activin signaling axis that is important for HCMV biology, to inhibit the activin A-triggered secretion of IL-6 in latently infected monocytes [104]. These data indicated that cellular receptors in immune cells are essential for HCMV-encoded miRNAs to reduce inflammatory responses and promote persistent infection. Other inflammatory pathways are also involved in HCMV-encoded miRNA-regulated cytokine secretion. A study by Hancock et al. showed that miR-UL112-3p and miR-US5-1 target the IκB kinase (IKK) complex components IKKα and IKKβ to reduce the secretion of proinflammatory cytokines in response to IL-1β and TNF-α [73,111]. In addition to the cell receptor pathway, HCMV-encoded miRNAs can also directly regulate the production of inflammatory factors. For example, HCMV-miR-UL112 regulates proinflammatory factors, including RANTES and IL-32, which are known to increase the cytolytic effect of NK cells. CD8^+^ T cells, epithelial cells, fibroblasts, and platelets release CCL5 after TNF-α and IFN-γ stimulation to recruit circulating leukocytes to viral infection sites. Unfortunately, miR-UL148D directly interacts with the CCL5 3′-UTR to induce its degradation for immune escape [77]. In summary, the delayed secretion of proinflammatory cytokines is one immune evasion mechanism exploited by virus miRNAs to limit the recruitment of cytocidal immune cells.

Hook L et al. [78] found that miR-US5-1, miR-US5-2, and miR-UL112-1 target multiple components of the host, including Vamp3, Rab5c, Rab11a, SNAP23, and CDC42, which are closely related to the secretory pathway. Due to the inhibited expression of these proteins, the three HCMV-encoded miRNAs effectively reduced the secretion of cytokines TNF-α and IL-6 in HCMV-infected cells, which could facilitate the formation of the virion assembly compartment (vAC) for efficient production of infectious virus. In summary, these data indicate that HCMV-encoded miRNAs maintain the persistence of latent infection by reducing inflammatory responses.

### 3.4. HCMV-Encoded miRNAs Induce Host Myelosuppression

HCMV latency is established when infected monocytes traffic to the bone marrow and spread infection to a small number of CD34^+^ hematopoietic progenitor cells (HPCs) [112]. This will cause hematopoietic dysfunction with major clinical manifestations, including myelosuppression, pancytopenia, bone marrow hypoplasia, and even transplant failure. However, the underlying mechanism that leads to the discrepancy between HCMV-loaded HPCs and myelosuppression has not been fully revealed. TGF-β has been acknowledged to regulate multiple cellular processes, such as cell survival, proliferation, differentiation, and migration [113]. TGF-β is also important for progenitor cell self-renewal and maintaining HPC homeostasis through the regulation of cytokine receptors, cell cycle regulators, or apoptosis [113]. Interestingly, the fluctuation of TGF-β concentration in the bone marrow leads to different observations that the presence of high TGF-β levels keeps HPCs dormant, while lower doses of TGF-β stimulate them [114]. According to a recent study, a latent HCMV infection of CD34^+^ HPCs induced the secretion of TGF-β, which is responsible for virus-mediated myelosuppression. HCMV encodes miR-US5-2 to downregulate the transcriptional repressor NGFI-A binding protein (NAB1) to achieve the myelosuppression of uninfected CD34^+^ HPCs. It also demonstrated that HCMV-encoded miR-UL22A blocked the TGF-β pathway by targeting SMAD3 in CD34^+^ HPCs with latent infection. These results provide a possible explanation for how a minor population of infected CD34^+^ HPCs induced overall myelosuppression. Taken together, HCMV-encoded miRNAs induce host myelosuppression and maintain lifelong infection in myeloid lineage cells at least partially through the TGF-β signaling pathway [79].

### 3.5. Host miRNAs Regulation toward Viral Genes

Changes in cellular miRNA expression also help to maintain latent HCMV infection. A study conducted by O’Connor and colleagues showed that HCMV deregulates the expression of viral UL122 (encoding IE2 protein) by cellular miRNAs to maintain latency [115]. This IE2 suppression could result in a decrease in several viral proteins (such as HCMV DNA polymerase or glycoproteins), which inhibit the release of complete HCMV virions [116]. They found that the viral titers in HCMV-infected fibroblasts decreased when human (hsa)-miR-200 was ectopically expressed. The hsa-miR-200 miRNA family members targeted the 3’-UTR of UL122 to prevent their translation and ensure that no reactivation occurs in myeloid cells unless they are differentiated. Moreover, they observed higher levels of hsa-miR-200 family members in cells (such as CD34^+^ hematopoietic progenitor cells (HPCs) and Kasumi-3 cells) where the virus favors latency compared with the cells that are permissive for reactivation, which was consistent with the data of Asou and his team [117]. They treated the CD34^+^ progenitor Kasumi-3 cell line with tetradecanoyl phorbol acetate (TPA) to induce their differentiation into the macrophage lineage and then compared the levels of the hsa-miR-200 cluster in untreated cells with those in treated cells. The results revealed that the hsa-miR-200 cluster was highly expressed in untreated Kasumi-3 cells but was significantly decreased after TPA-induced differentiation. Histone deacetylase 4 (HDAC4) is a class II HDAC that inhibits MIEP activity in myeloid cells. Krishna and his team found that HDAC4 is elevated in infected monocytes during latency, but its regulator hsa-miR-206 declines [118]. He et al. [119] reported that hsa-miR-182 targets FOXO3 to upregulate interferon regulatory Factor 7 (IRF7) expression and subsequently activates IFN-I to inhibit HCMV replication. These observations suggest roles for miRNAs oriented from host regulate vice versa to HCMV genes and particles to achieve virus latency.

## 4. Indispensable Role of HCMV-Encoded miRNAs in HCMV Reactivation

HCMV infects most organs and tissues, but its replication mechanisms significantly differ among infected host cells. Latent infection often occurs in less-differentiated cells such as CD14^+^ monocytes, CD34^+^ HPCs, and immature dendritic cells as mentioned above [16,120]. It has been recently reported that HCMV expresses only a subset of its miRNAs during latency [120]. The majority are originated from the UL region (e.g., miR-UL112, miR-UL36, and mir-UL22A) and few were from the US region (e.g., miR-US29-3p, and miR-US22-5p and miR-US5-1-5p) with unidentified regulatory mechanisms [120]. Some studies demonstrated that HCMV-encoded miRNAs are tissue-specific [121]. In fact, the expression levels of some HCMV-encoded miRNAs were confirmed to be dynamically changed in the same tissue during different stages of infection [122]. For example, the switch occurs upon myeloid precursor cell differentiation after external stimuli interaction (e.g., proinflammatory cytokines). These stimuli can alter intracellular environment in a manner that enables re-expression of viral IE, early (E), and late (L) gene products and lead to HCMV reactivation [123]. However, the substances and related signaling pathways are largely unknown [95].

HCMV reactivation can occur in immunocompromised populations to cause severe, sometimes life-threatening complications [124]. Virus particles in transplant recipients are mainly stored in CD34^+^ HPCs when the viral genome can be detected in CD34^+^ HPCs, as well as in peripheral blood monocytes. The expression of viral genes was restored during the HPC mobilization to the periphery and then the differentiation into macrophages or DCs [45,125]. Early growth response gene 1 (EGR-1) is an immediate early transcription factor located downstream of MEK/ERK signaling. It promotes CD34^+^ HPC self-renewal in the bone marrow niche and is downregulated when CD34^+^ HPCs mobilize from the bone marrow to the periphery [126]. Mikell et al. [83] proved that a binding site exists in the miR-US22 seed sequence for the EGR-1 mRNA 3′-UTR. In another experiment, HCMV miRNA mimics were cotransfected with an EGR-1 luciferase reporter into HEK293 cells with or without EGF stimulation. They found that luciferase expression by the EGR-1 reporter decreased 3- to 4-fold in miR-US22-transfected cells compared to the negative control, whereas the miR-US22 mutant failed to reactivate CD34^+^ HPCs. MiR-US22 is not expressed during HCMV latency in CD34^+^ HPCs, allowing the maintenance of the stem cell phenotype [79]. Upon miR-US22 upregulation, EGR-1 downregulation significantly blocked HPC self-renewal, which, in turn, induced their differentiation into the myeloid lineage. In a report by Hancock et al. [127], miR-US5-2 attenuates EGF-mediated mitogenic signaling by inhibiting Grb2-associated binding protein 1 (GAB1, the EGFR adaptor protein), which in turn reduces EGR-1 expression and cell proliferation. Ras homolog family member A (RhoA) is a Rho family GTPase critical for regulating actin dynamics, which is required for CD34^+^ HPC self-renewal and proliferation. Diggins et al. [128] found that miR-US25-1 deregulates RhoA, attenuates downstream signaling through myosin light chain II, and suppresses CD34^+^ HPC proliferation. All this evidence suggests that HCMV-encoded miRNAs and HPCs play essential roles in HCMV reactivation.

## 5. Conclusions and Outlook

HCMV is prevalent worldwide and causes lifelong latent infection, while it mainly contributes to the adverse prognosis of immunosuppressed patients upon reactivation. Due to minimal viral transcripts expressed, there are some troubles for technicians to raise a first alert to infection using present detection techniques; when HCMV is reactivated, it is intractable for doctors to assess cytomegalovirus disease stages due to its rapid progression and the suboptimal therapeutic choices thus far. It is far from enough to only detect nucleic acids for negative and positive results. Therefore, a comprehensive understanding of the HCMV-dominant molecular mechanisms is the key to prevention and treatment.

The development of HCMV vaccines has been going on for many years, some of which utilize viral protein antigens to elicit immune response, such as glycoprotein B/microfluidized adjuvant 59 (gB/MF59) vaccines, transgenic disabled infection single-cycle (DISC) vaccines, enveloped virus-like particle (eVLP) gB HCMV vaccines. There are also DNA-based HCMV vaccines (encoding both gB and pp65) which being conducted in preclinical or clinical trials and have achieved promising results [129]. Previous studies have proposed the insertion of miR-let-7b or miR-93 target sequences into the genome of influenza virus strain H1N1 or H5N1 in order to attenuate the virus [130,131]. In this review, we highlighted a series of HCMV-encoded miRNAs which are explicitly involved in HCMV reactivation (e.g., miR-UL148D) or in both latency and reactivation (e.g., miR-UL112). Therefore, development strategies of small molecule drug, as suggested by Abdalla [4], will be more directed and they are ideal for congenital infection intervene and cytomegalic inclusion disease (CID) treatment. In the meantime, multiple concomitant regulatory factors (e.g., IL-6 and TNF-α) are also underlined. Step-by-step validation in clinical laboratory is urgent, and solid experimental data need to be further analyzed for their promising value in clinical application [132].

## Figures and Tables

**Figure 1 vaccines-10-00144-f001:**
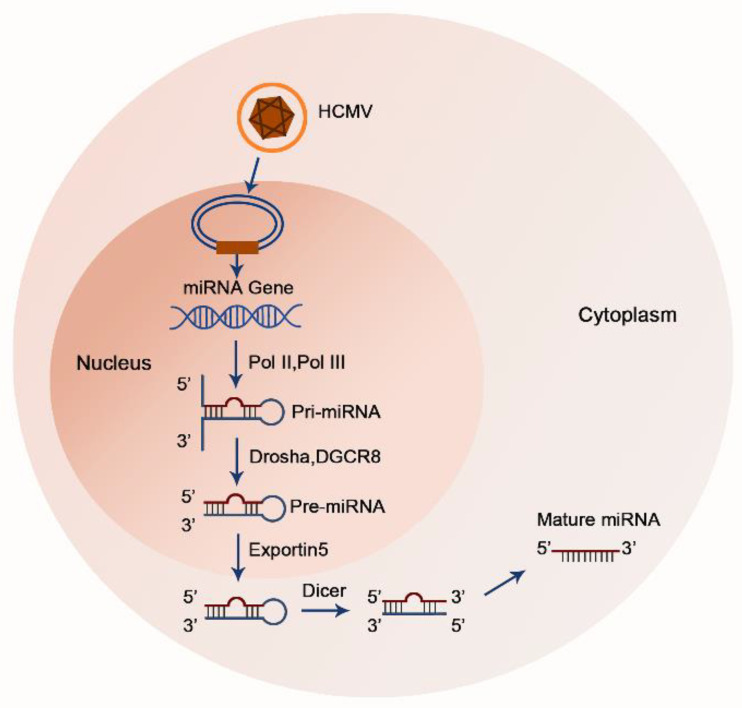
A sketch of HCMV-encoded miRNA biosynthesis model. Mature miRNAs are generated from hairpin secondary structures that arise from longer RNA polymerase II or polymerase III transcripts. In the nucleus, primary (pri-) miRNAs are cleaved into precursor (pre-) miRNAs via the microprocessor complex, consisting of DGCR8 and the ribonuclease Drosha. Next, pre-miRNAs are transported from the nucleus to the cytoplasm. After reaching the cytoplasm, pre-miRNAs are recognized and processed into their mature form by another RNase III, Dicer.

**Figure 2 vaccines-10-00144-f002:**
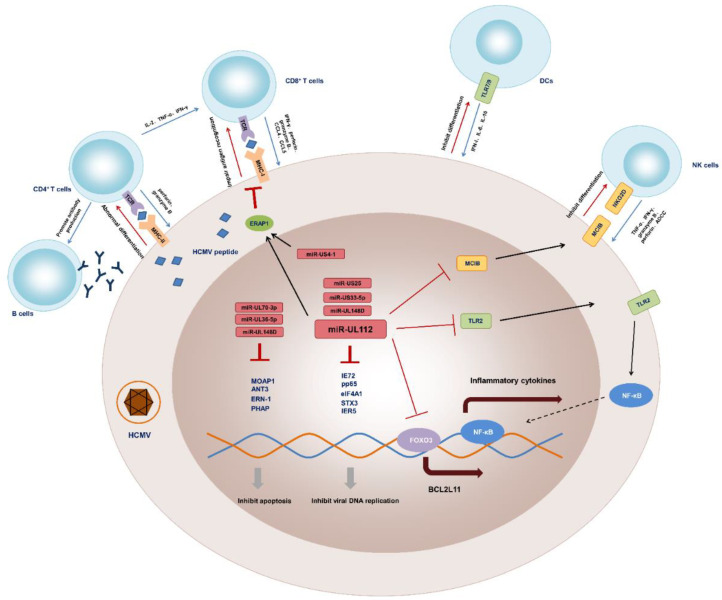
The interaction between HCMV-infected cells and immune cells. Innate immune response to early HCMV infection is mediated by APCs including DCs, NK cells through releasing multiple inflammatory cytokines. For example, HCMV-infected cells can directly trigger DCs to enhance antiviral IFN-I and secrete IL-6 and IL-10 through engagement of the TLR7 and/or TLR9 pathways. Activated NK cells promote anti-HCMV effect through binding NKG2D with its ligands to produce proinflammatory cytokines (e.g., TNF-α and IFN-γ) and cytotoxic granules containing effector molecules (e.g., granzyme B and perforin) to cause lysis or apoptosis of infected cells. NK cells also directly kill transformed and infected cells via antibody-dependent cellular cytotoxicity (ADCC). Besides, the control of HCMV viral replication and viral spreading is mediated by adaptive immune responses. For instance, CD4^+^ and CD8^+^ T cells recognize viral peptides presented by MHC-II and MHC-I molecules, respectively. Activated CD4^+^ T cells secret large amounts of IL-2, IFN-γ and TNF-α to promote the killing effect of CD8^+^ T cells and to assist B cells in producing antibodies. CD8^+^ T cells lyse infected cells through secreting granzyme B, perforin and IFN-γ. CD8^+^ T cells also produce CCL4 and CCL5 to recruit other immune cells to amplify local inflammation. Conversely, HCMV presents a series of miRNA-mediated strategies to resist immune attack and realize its latent infection. We take miR-UL112 as a typical example. In brief, miR-UL112 silences MICB mRNA translation thereby decreasing MICB binding to NKG2D and inhibiting differentiation of NK cells. Additionally, it attenuates NK cell-mediated cytotoxicity by downregulating inflammatory cytokines such as IL-32 and IFN, and TLR2-mediated NF-κB signaling. MiR-UL112-5p targets ERAP1, thereby inhibiting the processing and presentation of the HCMV pp65495-503 peptide to specific CTLs, and acts to reduce FOXO3 activity and BCL2L11 expression to promote infected cells’ survival. MiR-UL112 and other HCMV-encoded miRNAs also play essential roles in inhibiting viral DNA replication and apoptosis of host cells.

**Table 1 vaccines-10-00144-t001:** Summary of HCMV-encoded miRNAs.

Infection Stage	Function	HCMV-Encoded miRNA	Targets	References
Latency	Limit viral gene expression	miR-UL112-3p	HCMV IE72	[4]
			HCMV UL112/113	[55]
			HCMV UL120/121	
			UL114	[56]
		miR-UL148D	IER5	[57]
		miR-US25-1-5p	Cyclin E2	[58]
			TRIM28	[58]
			EID1	[58]
			MAPRE2	[58]
		miR-US25-2-3p	eIF4A1	[59]
		miR-US33-5p	CCND1	[60]
			STX3	[61]
	Escape immune response	miR-UL112-3p	MICA; NK cells	[62]
			MICB; NK cells	[63]
			IRF1; innate immune cells	[64]
		miR-UL112-5p	ERAP1; CD8^+^ T cells	[65]
		miR-US4-5p	ERAP1; CD8^+^ T cells	[47]
		miR-UL59	ULBP1; NK cells	[66]
		miR-US5-1	HCMV US7; multiple immune cells	[67]
		miR-US5-2-3p	HCMV US7; multiple immune cells	[67]
		miR-US33as-5p	IFNAR1; innate immune cells	[68]
	Inhibit autophagy	miR-UL112-3p	ATG5; HFFs	[61]
		miR-US22-5p	ATG5; HFFs	[61]
		miR-US29-5p	ATG5; HFFs	[61]
	Inhibit apoptosis	miR-US4-5p	CASP2; HFFs	[61]
		miR-UL112-5p	CASP3; HFFs	[61]
		miR-UL22A-5p	CASP3; HFFs	[61]
		miR-US25-2-3p	CASP3; HFFs	[61]
		miR-UL148D	IEX-1; HEK293 cells	[69]
			PHAP1; HeLa cell S-100	[70]
			ERN1; HeLa cell S-100	[70]
		miR-UL22A-3p	CASP7; HFFs	[61]
		miR-UL36-3p	FAS; HFFs	[61]
		miR-US5-1	FAS; HFFs	[61]
		miR-US5-2-3p	FAS; HFFs	[61]
		miR-UL36-5p	SLC25A6 (ANT3); HEK293 cells, U373 cells and HELF cells	[71]
		miR-UL70-3p	MOAP1; HEK293T cells	[71]
		miR-US4-5p	QARS; CD8^+^ T cells	[47]
		miR-US22-5p	US22; human fibroblast cells	[72]
	Reduce inflammatory cytokine production	miR-UL112-3p	IKKα/IKKβ; fibroblasts	[73]
		miR-US5-1	IKKα/IKKβ; fibroblasts	[73]
		miR-UL112-3p	IL-32; NK cells	[16]
			TLR2; NK cells	[74]
		miR-UL112-3p	Vamp3; NK cells	[75]
		miR-US5-1	Vamp3; NK cells	[75]
		miR-US5-2-3p	Vamp3; NK cells	[75]
		miR-UL112-3p	Rab5c; NK cells	[76]
		miR-US5-1	Rab5c; NK cells	[76]
		miR-US5-2-3p	Rab5c; NK cells	[76]
		miR-UL112-3p	Rab11a; NK cells	[75]
		miR-US5-1	Rab11a; NK cells	[75]
		miR-US5-2-3p	Rab11a; NK cells	[75]
		miR-UL112-3p	SNAP23; NK cells	[75]
		miR-US5-1	SNAP23; NK cells	[75]
		miR-US5-2-3p	SNAP23; NK cells	[75]
		miR-UL112-3p	CDC42; NK cells	[75]
		miR-US5-1	CDC42; NK cells	[75]
		miR-US5-2-3p	CDC42; NK cells	[75]
		miR-UL148D	ACVR1B; NK cells	[76]
			RANTES; NK cells	[77]
		miR-US25-1-5p	CD147; HEK293 cells	[78]
	Suppress cell cycle progression	miR-UL36-3p	CDK6; HFFs	[61]
		miR-US5-1	CDK6; HFFs	[61]
		miR-US5-2-3p	CDK6; HFFs	[61]
		miR-US25-1-3p	CDK6; HFFs	[61]
		miR-US25-2-3p	CDK6; HFFs	[61]
	Induce myelosuppression	miR-UL22A-3p	SMAD3; CD34^+^ HPCs	[79]
		miR-UL22A-5p	SMAD3; CD34^+^ HPCs	[79]
		miR-US5-2-3p	NAB1; CD34^+^ HPCs	[79]
Reactivation	Promote viral gene expression	miR-UL112-3p	BclAF1	[80]
		miR-UL36-5p	HCMV UL138	[81]
		miR-US5-1	Geminin	[82]
	Induce cell differentiation	miR-US22-5p	EGR1; HEK293 cells, NHDF	[83]
	Promote apoptosis	miR-US4-5p	PAK2; HEK293, HELF and THP-1 cells	[84]
		miR-US25-1-5p	BRCC3; EAhy926 cells	[85]
Others	N	miR-UL22A-5p	BMPR2	[86]
		miR-US4-3p	CASP7	[47]
			CDK6	[61]
			ERAP1	[47]
		miR-US22-3p	US22	[72]
		miR-US33-3p	US29	[87]
		miR-UL69	N	
		miR-UL70-5p	N	
		miR-US5-2-5p	N	
		miR-US25-2-5p	N	
		miR-US29-3p	N	

N = No targets or no exact function have been identified to date.

## Data Availability

Not applicable in a review article.

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
