# Peer review of "microRNA, a Subtle Indicator of Human Cytomegalovirus against Host Immune Cells"

_vaccines, 2022, doi:10.3390/vaccines10020144_

Round 1

Reviewer 1 Report

This reviewer has no major concerns about this review manuscript, and the following are some minor points for authors to improve their paper.

1) Figures are in low resolutions, and the characters are too small to be recognized clearly, e.g. Figure 2.

2) Although this review focuses on the host-virus interactions and the authors summarized thoroughly, it seems not such significant regarding how those mechanisms could direct future vaccine development. Another point is the current status of vaccines available in the clinical trial or other stages.

3) It is better to rearrange the references to include more recent literature since 2018.

Author Response

Response to Reviewer 1 Comments

Point 1: Figures are in low resolutions, and the characters are too small to be recognized clearly, e.g. Figure 2.

Response 1: Thank you for rising this problem. We have uploaded Figure 1 and Figure 2 with higher resolutions and larger characters in the revised manuscript.

Point 2: Although this review focuses on the host-virus interactions and the authors summarized thoroughly, it seems not such significant regarding how those mechanisms could direct future vaccine development. Another point is the current status of vaccines available in the clinical trial or other stages.

Response 2: Thank you for the helpful suggestion. The related content has been added to the “Conclusions and Outlook” of the revised manuscript (Line531-543).

Point 3: It is better to rearrange the references to include more recent literature since 2018.

Response 3: Thank you for the thoughtful comment. The main inclusion criteria of listed references were: 1) one had relevant contents as well as higher quality (e.g., sources, influence factor); 2) if detailed informations were described, the sentence was annotated with the original document (especially references in Table 1). We also have updated more recent literature since 2018 in the reviesed manuscript.

Reviewer 2 Report

The article “MicroRNA, A Subtle Indicator of Human Cytomegalovirus 2 against Host Immune Cells” is a very important contribution to the literature.

It is well written, provides a logical order of scientific data related to the subject and is properly organized.

I suggest a few details to enhance manuscript:

  1. Abstract: HCV was not preceded by its full description. (L-17)
  2. Figure 2 is essentially unreadable. It must be bigger.

Author Response

Response to Reviewer 2 Comments

Point 1: Abstract: HCV was not preceded by its full description. (L-17)

Response 1: Thank you for your helpful suggestion. The full description of HCV has been added in Line18 in the revised manuscript.

Point 2: Figure 2 is essentially unreadable. It must be bigger.

Response 2: Thank you for the valuable comment. We have uploaded Figure 1 and Figure 2 with higher resolutions and larger characters in the revised manuscript.

Reviewer 3 Report

I think it is a well put together and interesting review. However, it needs some revisions in order to be published.

As several review articles have been published recently, it is necessary to put forward novel points. In particular, since the findings are summarized with a focus on immune system regulation, it would be good to add the following points.

An explanation of miRNA production from viral genomes in cells is needed. In addition, a description of expression regulation is desirable. What is the stimulus for transcription? Is each miRNA transcriptionally regulated separately? Or is it regulated in metabolism, such as cleavage or complex formation, so that readers can more easily understand the regulation in the latent and acute phases.

Regarding Figure 2, the resolution of the figure is too low for me to understand.

The review by Abdalla only summarizes the conceptual methods of drug discovery. In this review, I think it would be more interesting if the authors explained the drug discovery strategies with more specific target diseases.

Throughout the review, the authors should clearly describe what the microRNAs they are describing target, and what MiR-UL70-3p in Line298 binds to in MOAP1 (e.g. 3'-UTR of mRNA) to regulate protein expression. Similarly, the miRNA-UL70-3p binding to Line271 should be clearly described. Similarly, in Line 271, miRNA-US25-1 and US25-2 are described as blocking expression, but this is an ambiguous expression.
It would be better to clearly state whether the translation is blocked, localized, or transported. Please check the entire description.

Line38; IE is the first time it appears as an abbreviation. The official name should be listed here.

Line67;Considering the contents of Figure 1, the appropriate citation is around line 59. miRNA DNA sequences are described as being transcribed by polymerase II, but is it only polymerase II that is involved?

Table 1 is a good reference, but it would be better if there were a table for each cell type that is affected by miRNAs. As for Table 1, since it is divided into infection stages, it would be better to describe the types of cells involved in each stage and clarify the role of miRNAs. In addition, it would be helpful to summarize the phenomena induced by miRNAs to facilitate overall understanding. 

RNA Biol 2021 18:2194-2202. Raquel Fernández-Moreno et al. should also be cited.

Author Response

Response to Reviewer 3 Comments

Point 1: An explanation of miRNA production from viral genomes in cells is needed. In addition, a description of expression regulation is desirable. What is the stimulus for transcription? Is each miRNA transcriptionally regulated separately? Or is it regulated in metabolism, such as cleavage or complex formation, so that readers can more easily understand the regulation in the latent and acute phases.

Response 1: Thank you for the valuable comments. The related contents have been added in Line68-71, Line74-78, and Line480-494 in the revised manuscript.

Point 2: Regarding Figure 2, the resolution of the figure is too low for me to understand.

Response 2: Thank you for your helpful suggestion. We have uploaded Figure 1 and Figure 2 with higher resolutions and larger characters in the revised manuscript.

Point 3: The review by Abdalla only summarizes the conceptual methods of drug discovery. In this review, I think it would be more interesting if the authors explained the drug discovery strategies with more specific target diseases.

Response 3: Thank you for the thoughtful comment. Detailed informations have been added in Line531-543 in the revised manuscript.

Point 4: Throughout t he review, the authors should clearly describe what the microRNAs they are describing target, and what MiR-UL70-3p in Line298 binds to in MOAP1 (e.g. 3'-UTR of mRNA) to regulate protein expression. Similarly, the miRNA-UL70-3p binding to Line271 should be clearly described. Similarly, in Line 271, miRNA-US25-1 and US25-2 are described as blocking expression, but this is an ambiguous expression.

It would be better to clearly state whether the translation is blocked, localized, or transported. Please check the entire description.

Response 4: Thank you for the valuable comments. We have marked all the changes in red in the revised manuscript.

Point 5: Line38; IE is the first time it appears as an abbreviation. The official name should be listed here.

Response 5: Thank you for your helpful suggestion. The official name of IE has been added in Line39 in the revised manuscript.

Point 6: Line67;Considering the contents of Figure 1, the appropriate citation is around line 59. miRNA DNA sequences are described as being transcribed by polymerase II, but is it only polymerase II that is involved?

Response 6: We accepted the valuable suggestion. We have marked the changes in red in Line60 and in Figure 1 legend in the revised manuscript.

Point 7: Table 1 is a good reference, but it would be better if there were a table for each cell type that is affected by miRNAs. As for Table 1, since it is divided into infection stages, it would be better to describe the types of cells involved in each stage and clarify the role of miRNAs. In addition, it would be helpful to summarize the phenomena induced by miRNAs to facilitate overall understanding.

Response 7: The related contents has been added in Table 1 in the revised manuscript. We can list these contents in another Table, if you think it would be better.

Point 8: RNA Biol 2021 18:2194-2202. Raquel Fernández-Moreno et al. should also be cited.

Response 8: The article has been cited in Reference 132 in the revised manuscript.

Round 2

Reviewer 3 Report

I recommend the accept after Minor revision. Size of characters in  Fig 2 is too small. Please change.